# Partial Agonistic Actions of Sex Hormone Steroids on TRPM3 Function

**DOI:** 10.3390/ijms222413652

**Published:** 2021-12-20

**Authors:** Eleonora Persoons, Sara Kerselaers, Thomas Voets, Joris Vriens, Katharina Held

**Affiliations:** 1Laboratory of Endometrium, Endometriosis & Reproductive Medicine, Department of Development and Regeneration, KU Leuven, Herestraat 49 Box 611, 3000 Leuven, Belgium; eleonora.persoons@kuleuven.be (E.P.); kathi.held@kuleuven.be (K.H.); 2Laboratory of Ion Channel Research, Department of Cellular and Molecular Medicine, KU Leuven, VIB Center for Brain & Disease Research, Herestraat 49 Box 802, 3000 Leuven, Belgium; sara.kerselaers@kuleuven.be (S.K.); thomas.voets@kuleuven.be (T.V.)

**Keywords:** transient receptor potential channel, TRPM3, steroid, DHEAS, estradiol, progesterone, testosterone

## Abstract

Sex hormone steroidal drugs were reported to have modulating actions on the ion channel TRPM3. Pregnenolone sulphate (PS) presents the most potent known endogenous chemical agonist of TRPM3 and affects several gating modes of the channel. These includes a synergistic action of PS and high temperatures on channel opening and the PS-induced opening of a noncanonical pore in the presence of other TRPM3 modulators. Moreover, human TRPM3 variants associated with neurodevelopmental disease exhibit an increased sensitivity for PS. However, other steroidal sex hormones were reported to influence TRPM3 functions with activating or inhibiting capacity. Here, we aimed to answer how DHEAS, estradiol, progesterone and testosterone act on the various modes of TRPM3 function in the wild-type channel and two-channel variants associated with human disease. By means of calcium imaging and whole-cell patch clamp experiments, we revealed that all four drugs are weak TRPM3 agonists that share a common steroidal interaction site. Furthermore, they exhibit increased activity on TRPM3 at physiological temperatures and in channels that carry disease-associated mutations. Finally, all steroids are able to open the noncanonical pore in wild-type and DHEAS also in mutant TRPM3. Collectively, our data provide new valuable insights in TRPM3 gating, structure-function relationships and ligand sensitivity.

## 1. Introduction

Sex hormones control several crucial body functions as they act on multiple different tissues in our endocrine and nervous system. These actions occur primarily via the classical genomic pathway, which requires the involvement of nuclear receptors that intracellularly bind these sex hormones and subsequently affect gene transcription mechanisms [1,2]. However, (precursor) sex hormones were also found to bind to membrane receptors, thereby modulating other signaling cascades [3,4]. One family of proteins comprising such membrane receptors is the superfamily of transient receptor potential (TRP) channels. This class of cationic permeable ion channels are membrane-spanning proteins that are responsive to a diverse range of stimuli including physical, thermal as well as chemical cues. In fact, it has been already reported that some members are modulated by steroid sex hormones, including TRPM8 by testosterone, TRPA1 and TRPV1 by androstenedione [5,6,7], as well as TRPM3 by a limited number of steroids [8].

Over the last decade, the interest in TRPM3 has strongly increased, as research has discovered its role in insulin-secretion, vascular contraction, heat sensing, inflammatory hyperalgesia and spontaneous pain [8,9,10,11,12,13,14,15]. TRPM3 is a nonselective calcium-permeable cation channel that is polymodally gated by both heat and chemical compounds. The best-known endogenous chemical agonist of TRPM3 is the neurosteroid pregnenolone sulphate (PS), a derivative of cholesterol (Figure 1), which induces rapid and reversible activation of the ion channel [8]. Other ligands that cause increased TRPM3 channel activity are the dihydropyridine nifedipine and the synthetic small molecule CIM0216 [8,14].

As cholesterol also gives rise to other structurally related (precursor) sex hormones, like dehydroepiandrosterone sulphate (DHEAS), progesterone (P_4_), testosterone (T), and estradiol (E_2_), and TRPM3 is shown to be highly expressed in gonads [16,17], researchers became curious to explore whether these hormones have a modulatory effect on TRPM3 channel activity. For example, Wagner et al. tested a wide array of steroids including DHEAS, progesterone, and testosterone [8]. However, their research only identified DHEAS as a TRPM3 agonist at a concentration of 50 µM, albeit a less potent one than PS. Moreover, other research states that progesterone and estradiol can even act as inhibitors of TRPM3 [18].

Interestingly, several new insights in the working mechanisms of TRPM3 have been discovered in recent years, but actions of PS-related steroid hormones have not been further considered. For instance, increasing temperature to physiological levels, i.e., 37 °C, is known to strongly sensitize TRPM3 for PS, leading to synergistic actions [10]. However, literature reporting steroidal actions on TRPM3 does not take the heat sensitivity of the channel into account, as this was discovered only at a later stage [10,13,19]. Furthermore, recent research on TRPM3 described the existence of a noncanonical ion permeation pathway that is predicted to be located in the voltage-sensing domain of the channel and shown to induce strong depolarization when opened [20,21,22]. Opening of this pathway occurs upon simultaneous application of the endogenous ligand PS and the antifungal drug clotrimazole [20] as well as by sole application of the synthetic TRPM3 agonist, CIM0216 [14]. However, the impact of other PS-related sex steroids on the noncanonical pore remains to be elucidated. [20]. Finally, several studies have recently described that patients with TRPM3-mutations can have distinct morphological features and suffer from symptoms such as intellectual disability, speech retardation, epileptic seizures or heat insensitivity [23,24,25,26]. Further in-depth characterization has shown that two human disease variants in TRPM3 (V990/2M and P1090/2Q), which result in developmental and epileptic encephalopathy (DEE), can cause increased responses to PS, leading to elevated cytosolic calcium levels compared to wild-type TRPM3 [27,28]. For the V990M variant, this is partially caused by the ability of PS to open the noncanonical ion permeation pathway in the absence of the comodulator clotrimazole. Apart from PS, no other steroid was investigated concerning its actions on TRPM3 DEE mutations, although several of them are reported to be present in the brain [29].

In this work, we aim to clarify and deepen our understanding of sex-steroidal actions on TRPM3 gating in wild-type and disease-associated variant channels by means of calcium imaging and patch clamp recordings.

## 2. Results

### 2.1. DHEAS, E_2_, P_4_ and Testosterone Act as Weak Agonists of TRPM3 at Body Temperature

To start, calcium-microfluorimetric experiments were performed at 23 °C and 37 °C on either HEK-cells stably expressing murine TRPM3α2 (HEK-TRPM3) or nontransfected HEK293T (NT) cells, to investigate the potential modulatory effect of DHEAS, E_2_, P_4_ and T on TRPM3 at body core temperature. At 23 °C, 300 µM DHEAS and E2 could elicit a moderate increase in [Ca^2+^]_i_ (Figure 2A,C) while only a limited rise in intracellular calcium levels was observed for 300 µM P4 or T (Figure 2E,G). When these experiments were repeated at body core temperature, the responses were clearly enhanced. In case of 300 µM DHEAS at 37 °C, a robust and reversible increase of the intracellular calcium levels was observed, which was not detected in NT cells (Figure 2A). Next, concentration-response curves of DHEAS were measured, using calcium imaging on HEK-TRPM3 cells at both room (23 °C) and body temperature (37 °C) (Figure 2B). Indeed, a leftward shift of the concentration-response curve was observed. Likewise, a Ca^2+^ influx was measured when HEK-TRPM3 cells were exposed to 300 µM E_2_, while NT cells did not show this influx (Figure 2C). Concentration-response curves showed again a noticeable shift of the curve after increasing the temperature to 37 °C (Figure 2D). Moreover, an application of 300 µM progesterone caused a rise in intracellular Ca^2+^ levels in HEK-TRPM3 cells, while NT cells did not show an increase in intracellular Ca^2+^ upon administration (Figure 2E). Similar to DHEAS and estradiol, a leftward shift of the concentration-response curve was observed at 37 °C (Figure 2F). Next, an administration of a high dose (300 µM) of testosterone also elicited a rise of intracellular Ca^2+^ levels in HEK-TRPM3 cells, which was not detected in NT cells (Figure 2G), with a leftward-shift of the concentration-response curve when performed at 37 °C (Figure 2H). Additionally, patch clamp data confirmed the agonistic action of the tested sex steroids, as an increase in TRPM3 currents was observed at room temperature upon application of these steroids (Appendix A). Although these novel results partially conflict with previous publications, it must be noted that of all tested steroids, DHEAS is the most potent, as was previously reported. Therefore, the agonistic potential and its temperature-sensitivity was also tested for DHEAS on the human orthologue. Concentration-response experiments on HEK-hTRPM3 cells at both, 23 °C and 37 °C, resulted in a similar leftward-shifted curve as was observed in HEK-TRPM3 cells (Appendix A).

The TRPM3-specificity of these steroids was further verified by the use of the TRPM3-specific antagonist isosakuranetin [30]. Calcium responses were largely abolished when the different steroids were applied in the presence of isosakuranetin (5 µM) (Figure 3). When isosakuranetin was subsequently removed in the continued presence of the steroids, a robust rise in [Ca^2+^]_i_ was noticeable for all steroids. Upon reapplication of the antagonist, this response was blocked, indicating that these steroid-induced increases in intracellular Ca^2+^ are inhibited by the TRPM3 antagonist (Figure 3).

### 2.2. Sex Steroid Hormones Share a Similar Interaction Site and Gating Features

Next, we investigated whether there is competition between the weaker agonists DHEAS, E_2_, P_4_ or T and the more potent PS. Indeed, calcium-imaging experiments showed that application of DHEAS, E_2_, P_4_ or T (100 µM) induced a partial and reversible block of the PS-induced calcium response by 8.0 ± 0.5%, 10.1 ± 0.3%, 8.5 ± 0.2%, and 9.6 ± 0.3% respectively. This steroid-induced block of the PS response was reversible upon washout of DHEAS, E_2_, P_4_ or T (Figure 4). Importantly, control recordings with the vehicle (0.1% DMSO) resulted only in a weak decrease of the PS-response by 1.1 ± 0.5% (Appendix A), confirming the antagonistic action of the steroidal drugs on PS-induced TRPM3 responses. Oppositely, DHEAS caused a small potentiation of the response to the nonsteroidal agonist nifedipine (Appendix A). In contrast to the difference in potency of the agonistic effect of the steroids, no difference was observed in the potency of inhibition. Collectively, these data strongly support the idea of a shared interaction site for sex hormone steroidal drugs on TRPM3.

Recent work from our group has identified an alternative ionic pathway in TRPM3 that is formed by the voltage sensing domain (S1–S4), which can be gated by combined application of PS and the exogenous chemical clotrimazole (Clt) [14,20]. This is exemplified by the strong potentiation of PS-induced outward and importantly inward currents in HEK-TRPM3 when a preapplication of Clt is given [20]. Here, we investigated whether this property is also observed when Clt was co-applied with the other steroids DHEAS, E_2_, P_4_ and T. Indeed, preincubation of Clt followed by a co-application of DHEAS and Clt resulted in robust outward and inward currents (Figure 5A,B). Most importantly, these currents presented a significant increase compared to the currents recorded during single application of DHEAS (Figure 5C). Similar results were obtained for estradiol, progesterone, and testosterone, whereby the fold-increases of the TRPM3-currents were in a similar range for progesterone and lower for estradiol and testosterone compared to DHEAS (Figure 5). Next, the synthetic drug, CIM0216, is reported to be able to open both the canonical and the noncanonical pore of TRPM3 [14]. Moreover, it was also shown that PS can further potentiate CIM0216-induced currents, in a similar manner as it potentiates currents in the presence of Clt [14]. Therefore, similar recordings were performed investigating the potentiating effect of steroids on CIM0216-induced currents. These experiments illustrated that all four steroids were able to induce a potentiation of the CIM0216-induced currents (Appendix A).

### 2.3. Sex Steroid Hormones May Contribute to the Disease Phenotype of Patients Carrying Trpm3 Gene Alterations

Recently, two independent studies reported that two TRPM3-variants associated with human neurodevelopmental disease, V990/2M and P1090/2Q, are gain of function variants exhibiting a higher sensitivity to PS and heat stimulation [27,28]. Since sex steroid hormones are known to play important roles in the brain [29], the agonistic potential of the here investigated steroids was further tested in both TRPM3 variants V990M and P1090Q. HEK293T cells were transiently transfected with the V990M or P1090Q mutant channel, respectively, and investigated for their sensitivity towards steroid stimulation. As was described earlier, both mutant channels show increased basal intracellular calcium levels (Figure 6A,C,E,G and Appendix A). Moreover, for the V990M mutant, Fura-2-based calcium imaging showed a significantly increased Δ[Ca^2+^]_i_ compared to the wild-type (WT) TRPM3 channel upon application of 100 µM DHEAS, E_2_ or 300 µM P_4_ or T (Figure 6). Interestingly, cells expressing the P1090Q mutant only showed an increased Δ[Ca^2+^]_i_ for two of the steroids, namely DHEAS and E_2_, (Figure 6A–D), while a reduction of the intracellular Ca^2+^ concentration was observed for P_4_ and T stimulation (Figure 6E–H). An increased Δ[Ca^2+^]_i_ upon treatment with the sex steroids suggests a higher sensitivity for the tested hormones, which is in line with the reported sensitization for PS in V990M and P1090Q. In order to confirm such sensitivity shift, concentration-response curves were measured in calcium-imaging experiments for DHEAS in WT and both TRPM3 disease-mutants. These results confirmed a sensitivity increase for DHEAS in both mutant channels, as proven by the leftward shift in the concentration-response curves compared to WT TRPM3 (Appendix A).

As earlier research accounted part of the sensitizing effect for PS in the DEE mutants to the fact that PS is able to open the noncanonical pore of TRPM3 in the absence of another ligand such as clotrimazole, additional experiments were performed investigating whether DHEAS and E_2_ can act in a similar manner. Patch clamp experiments were performed on V990M transfected cells, as in comparison to P1090Q it was shown to possess more pronounced inward currents that are representative for the opening of the noncanonical pore. Fittingly, the currents in the presence of DHEAS showed a prominent inwardly rectifying current component in the V990M variant, which was not observed in the WT channel, with a similar IV-shape as for the PS-evoked currents (Figure 7A,B). In case of estradiol, whole-cell currents lacked the characteristic inwardly rectifying component in both the wild-type channel and the V990M mutant (Figure 7C,D). However, both DHEAS and estradiol induced increased current densities (ΔI) in the V990M mutant compared to WT (Figure 7E), suggesting a sensitization of the mutant channel to steroid hormones. The absence of the characteristic inward currents in the estradiol treated V990M mutation, proposes that the sensitization of the mutant to estradiol is independent of the opening of the noncanonical pore. Indeed, when normalizing the IV-curves to the maximal current that was recorded in each cell at +150 mV, the V990M mutant showed similarly shaped normalized IV-curves in the presence of estradiol, as could be observed in WT in both treatment conditions (DHEAS and E_2_) (Figure 7F). However, the normalized currents were increased in amplitude compared to both treatment conditions in WT. Contrary, when treating the V990M mutant channel with DHEAS, clear outward and inward rectifying currents were observed with a similar amplitude increase of outward and inward currents. Collectively, these data support the conjecture that DHEAS and estradiol are exhibiting a higher potency in the V990M variant channel, whereas only DHEAS induces the gating of the noncanonical pore, which leads to strong cell depolarizations at physiological voltages.

## 3. Discussion

In this study, we investigated whether the (precursor) sex hormones DHEAS, E_2_, P_4_, and testosterone act on TRPM3 at physiological body core temperatures, considering the fact that TRPM3 emerged as a temperature-regulated ion channel that is highly expressed in gonads [10,13,16,17,19]. Moreover, we gained further insights into the effects of sex hormones on TRPM3 gating in order to better understand structure-function relationships of TRPM3 as well as possible structural requirements for ligands.

Fura-2-based calcium imaging experiments indicated that, at room temperature, only high concentrations (ranging from 1–300 µM) of the (precursor) steroids could activate TRPM3. This is in line with previous reports, where little to no activation was described at 23 °C for the sex hormones progesterone, estradiol, and testosterone [8,9,18]. Only the precursor sex hormone DHEAS was reported to have some agonistic potential at room temperature. Interestingly, experiments performed at body core temperature showed a leftward shift of the concentration-response curves. The concentration-dependent relationship identified DHEAS as the most potent tested steroid, followed by estradiol, testosterone, and progesterone. A potential explanation for the difference in potency among the different steroids could be the strong structural similarity between DHEAS and PS. In fact, the only difference between both steroids is the cyclopentane ring carrying an oxygen instead of an acetyl group (Figure 1) [31]. However, a similar rationale could count for progesterone, which is also closely related to PS, but lacks the sulphate group. Indeed, earlier studies already pointed out that the sulphate group at position C3 is of particular importance for the potency of TRPM3 activation by steroids [8,31,32].

Our results demonstrate, for the first time, that estradiol, progesterone and testosterone act as agonists of TRPM3. Indeed, the TRPM3-specificity of the observed agonistic responses was confirmed using the selective TRPM3 blocker isosakuranetin. These results are in discrepancy with previous reports. Literature has labelled progesterone and estradiol as inhibitors of TRPM3, due to their limited capability of stimulating TRPM3 and their ability to partially inhibit a PS-induced response [18]. However, most research on these steroids has been performed at room temperature, where the influx of ions through a noxious heat sensor like TRPM3 is vastly reduced. Furthermore, in the past, lower concentrations than in this manuscript were used when steroids, like progesterone, were tested at temperatures of 37 °C [18]. However, it should be noted that former and present experiments were performed in differing heterologous overexpression systems, which makes it difficult to predict whether the used concentrations are appropriate to imitate similar dynamic ranges of steroids and ion channels. Based on our obtained results, we suggest renaming all four tested hormones as partial/weak agonists, which compete with PS binding, as was indicated by the small block of the PS-induced Ca^2+^ influx that all four steroids are exerting in TRPM3 expressing cells (Figure 4). The fact that these neurosteroids can function as partial agonists of the nociceptor TRPM3, could suggest a potential role of neurosteroids in the regulation of nociceptive and neuropathic pain. For example, fluctuations in estrogen and progesterone in different phases of the menstrual cycle affect pain perception [33] and the nociceptive thresholds differ across the estrous cycle of rats as well [34]. Moreover, for DHEAS and testosterone it has been shown that these compounds could impact the spinal nociceptive transmission [35].

Interestingly, all tested steroids showed a blocking effect of the PS-response around ±10%. This could be explained, albeit not exclusively, by a shared interaction site between the two steroids. For DHEAS, we acquired more additional evidence to investigate this hypothesis. Namely, DHEAS increased the TRPM3-response in the presence of the non-steroidal drug, nifedipine, which likely binds to a different ligand-binding site than the steroids. The blocking effect of neurosteroids on the PS-induced Ca^2+^ influx could potentially explain the analgesic effect of spinal intrathecal administration of testosterone [36]. Moreover, earlier reports have shown that the individual differences in testosterone levels may be related to anti-nociception and protection against noxious stimuli in healthy women [37]. Additionally, testosterone has been reported to induce analgesia in male rats [38]. However, more in-depth research is needed to demonstrate whether the effect of neuro-steroids on TRPM3 can be linked to pain perception and whether the required concentrations are physiologically relevant. In summary, our results confirm agonistic actions of steroidal hormones on TRPM3, which were also already reported for other TRP channels before. For instance, testosterone can activate TRPM8 in the pico- to nanomolar range [5]. For P_4_ and E_2,_ no direct effect on other TRP channels has been described, although they are known to regulate the gene expression of TRPV1, TRPV5, TRPA1 and TRPM8 [7].

TRP channels can form tetramers, which will assemble to a single central cation-conducting pore that can be opened, for instance, upon the binding of ligands. Alternatively, for TRPM3, an alternative ion permeation pathway, located in the voltage-sensing domain, can be gated by either the combined application of PS and the chemical clotrimazole or by single application of the synthetic compound CIM0216 [14,20,21]. The opening of this noncanonical pore results in a marked potentiation of TRPM3-currents evoked by PS. Here, we elucidate that other sex hormones can also contribute to the gating of the noncanonical pore. Preincubation with the exogenous compound clotrimazole potentiated TRPM3 currents of all tested steroid hormones. The potentiation was the most pronounced for DHEAS and P_4_, but was also observed for E_2_ and T. Similarly, CIM0216-evoked currents were potentiated by the application of any of the four tested steroids, which aligns with the strong potentiation seen upon application of CIM0216 and PS [14]. Although both clotrimazole and CIM0216 are not endogenously expressed in the human or murine body, we cannot exclude, at this moment that potentially an endogenous compound exists that exhibits clotrimazole- or CIM0216-like properties, influencing TRPM3 gating behavior. Additionally, these data provide an extended explanation for the experienced side effects during Clt treatment and thereby helps in our understanding of the working mechanisms of a commonly used over-the-counter drug. Finally, our results might help in the development of future safe and secure TRPM3–specific drugs that are currently under investigation with regard to the nociceptive role of TRPM3.

Next, we explored the effect of these steroids on newly reported TRPM3 disease variants. De novo substitutions in the TRPM3 gene encoding the V990M and P1090Q variants have been observed in patients with intellectual disability and epilepsy, which was first described by Dyment et al. [23]. Subsequently, electrophysiological measurements were performed to investigate their impact on TRPM3 functionality [27,28]. For both mutants, a gain-of-function was observed, which includes an increased basal activity and a higher sensitivity to PS and heat. Moreover, the V990M variant exhibited a pronounced activation of the noncanonical currents through the voltage-sensor domain upon sole application of PS. Therefore, we were interested to examine whether patients with the recurrent mutant V990M and P1090Q show an increased sensitivity towards other steroid hormones. Fura-2-based calcium imaging experiments on the V990M mutant illustrated a significant increase in the TRPM3-response for all tested steroids in comparison with WT TRPM3. This is in line with a higher sensitivity of the mutant channel for steroid hormones, as was already published for PS. In contrast, the P1090Q mutant only showed a significant increase of the TRPM3-response towards DHEAS and E_2_ stimulation. The other steroids, P_4_ and T, inhibited TRPM3 channel activity. This might be attributed to the location of the mutation, as the substitution of proline by glutamine is occurring in the pore-loop of TRPM3, while the V990M mutation is located in the linker between transmembrane segments S4 and S5. It was shown in earlier reports that PS is interacting with TRPM3 from the extracellular side [8]. Therefore, it is possible that steroidal drugs are interacting in proximity to the Pro at position 1090 in contrast to the Val 990 that is located at the intracellular side of the channel. This could lead to varying effects of this mutation on steroidal actions depending on their chemical properties and/or sizes. However, further research is required to elucidate whether this hypothesis is correct. Inarguably, a TRPM3 structure combined with molecular dynamics simulations could help answer potential interactions of steroid hormones with TRPM3. However, such tools are currently not available.

Lastly, we investigated the ability of steroids to open the noncanonical pore in these mutants. A typical characteristic of the V990M mutant is that the opening of the noncanonical pore is enhanced, leading to larger inward currents as compared to WT. Therefore, the actions of the two most potent steroids, DHEAS and estradiol, were assessed on V990M. Patch-clamp data confirmed the increased sensitivity of the mutant for both steroids. However, when the currents were normalized to the maximum currents at +150 mV, a marked difference was noted between DHEAS and E_2_. In case of the former, a high similarity was observed between the values at −150 mV and +150 mV, as the current exhibited a prominent inwardly rectifying component typical for the opening of the noncanonical pore. In contrast, normalized estradiol responses showed distinct values at −150 mV and at +150 mV. A potential explanation for this could be that activation of the V990M mutant via E_2_ induces solely the gating of the central pore, whereas DHEAS induces the opening of both the central and noncanonical pores. All together, these data might provide new insights in the development and progression of DEE in patients with TRPM3-mutations. Several studies have described that patients with TRPM3-mutations can have distinct morphological features and suffer from symptoms such as intellectual disability, speech retardation, epileptic seizures or heat insensitivity [23,24,25,26]. During early childhood, steroid hormone levels are rather low. However, after birth, during the first months of life, a surge in testosterone and estradiol levels can be observed in boys and girls, respectively, known as ‘mini-puberty’. This event is thought to have a potential influence on body composition and growth, cognitive and language development and even behavior [39]. Increased Ca^2+^ influx caused by the sensitization of TRPM3 to these steroid hormones during this period might lie at the basis of the neurodevelopmental symptoms. However, further research is required to further investigate this hypothesis.

Despite the agonistic character of (precursor) sex hormones that we observed in this study, it must be noted that the used concentrations are supraphysiological, as their concentrations in the body are in the pico- to nanomolar range, with DHEAS being the most abundant circulating steroid in humans at only 1–10 μM. However, their mode of action might vary depending on space- and time-dependent changes in the hormonal levels. For instance, sex hormones are produced in the adrenal glands and the gonads [40,41], both tissue types where TRPM3 is also abundant [16,17,42,43]. Therefore, the agonistic effect of steroid hormones on TRPM3 might serve as feedback loop signal on these tissues, in order to fine-tune hormonal levels. In fact, feedback loops are a general feature of the endocrine system and employed at several levels during hormone expression, production, secretion and excretion. Moreover, based on the broad distribution of TRPM3 in the human body [44], various effects of steroidal compounds can be expected. Development, fertility, immunity, pain perception and metabolism, amongst others, might be stirred by actions that sex hormones exert on TRPM3. However, at this point, these are just speculations and more research is required in order to determine specific actions of steroids on TRPM3 in different tissues and how this might contribute to different physiological functions. Furthermore, it cannot be excluded that (precursor) sex hormones open the noncanonical pore, leading to massive cationic influxes through TRPM3 in certain (disease) states, either via an endogenous Clt-analogue or due to *Trpm3* gene alterations.

To conclude, we provided evidence that DHEAS, estradiol, progesterone, and testosterone are partial agonists of TRPM3 that can compete for the same binding site with the more potent pregnenolone sulphate. These steroids are able to activate the noncanonical pore. Notably, human disease-associated TRPM3 variants exhibit an increased sensitivity to these steroids.

## 4. Materials and Methods

### 4.1. Cell Culture

Nontransfected HEK293T (NT) cells, HEK-cells stably expressing murine TRPM3α2 (HEK-TRPM3) and HEK-cells containing the human TRPM3 coding region under the control of a tetracycline-regulated promoter (HEK-hTRPM3) were designed and cultured as previously described [27]. In the latter cells, expression was induced by adding 1 µg/mL tetracycline to the culture medium 24 h prior further experiments. As a control, cells that had not received tetracycline treatment (-Tetracycline HEK-hTRPM3) were used. HEK293T cells were transiently transfected with 2 µg of wild-type hTRPM3-YFP, V990M hTRPM3-YFP or P1090Q hTRPM3-YFP using TransIT transfection reagent (Mirus, Madison, WI, USA) 36–48 h before the measurements.

### 4.2. Calcium Imaging

The protocol and imaging system for standard Ca^2+^-measurements were similar as described in established protocols from our research group [10]. In brief, to image intracellular calcium, cells were incubated with 2 µM Fura-2 acetoxymethyl ester (Biotium, Fremont, CA, USA) at 37 °C for 30 min prior to the measurement. During the imaging protocol, cells were in an adapted Krebs’ solution (150 mM NaCl, 10 mM HEPES, 1 mM MgCl_2_, 2 mM CaCl_2_) where TRPM3 channel activity was investigated by the application of compounds, diluted in Krebs’, administered to the direct proximity of the recording field. Compounds entailed DHEAS (stock 100 mM), E_2_ (stock 100 mM), P_4_ (stock 100 mM), T (stock 100 mM), PS (100 mM stock), isosakuranetin (10 mM stock), clotrimazole (10 mM stock) and CIM0216 (10 mM stock). Fluorescent signals were evoked by alternating illuminations at 340 and 380 nm using a Lambda XL illuminator (Sutter instruments, Novato, CA, USA) and recorded using an Orca Flash 4.0 camera (Hamamatsu Photonics, Hamamatsu City, Japan) on a Nikon Eclipse Ti fluorescence microscope (Nikon, Tokyo, Japan). The imaging data were recorded and analyzed using the NIS-elements software (Nikon, Tokyo, Japan) and IgorPro (WaveMetrics, Portland, OR, USA). Absolute calcium concentrations were calculated from the ratio of the fluorescence signals at both wavelengths (F340/F380) after correction for the individual background fluorescence signals, using the Grynkiewicz equation [45]. Concentration–response curves were fitted with a function of the following form:(1)$$Y=A1+\frac{A2-A1}{1+{\left(\frac{V50}{x}\right)}^{h}}$$
where *A*1 is the bottom asymptote, *A*2 the top asymptote, *V*50 the half maximal effective concentration (EC50) or the half maximal inhibitory concentration (IC50) and h the hill coefficient [46]. The data were analyzed with IgorPro (WaveMetrics, Portland, OR, USA), OriginPro (OriginLab, Northhampton, MA, USA) and Prism (Graphpad, Jolla, CA, USA) software.

### 4.3. Patch Clamp Recordings

Whole-cell patch clamp recordings were performed as previously described [10,14]. Briefly, all recordings were done with an EPC-10 amplifier and the PatchMasterPro software (HEKA. Lambrecht, Germany). Currents were sampled at 20 kHz and digitally filtered at 2.9 kHz. Minimally 50% of the series resistance was compensated during the recordings. The standard internal solution contained (in mM): 100CsAsp, 45 CsCl, 10 EGTA, 10 HEPES, 1 MgCl_2_ (pH 7.2 with NaOH). The standard external solution contained (in mM): 150 NaCl, 1 MgCl_2_, 10 HEPES (pH 7.4 with NaOH). Cells were patched with a standard pipette resistance between 2–4 MΩ when filled with the internal solution. The standard recording voltages were applied as a 500 ms ramp protocol ranging from −150 mV to +150 mV. All experiments were performed at room temperature (23 ± 1 °C). The data were analyzed with IgorPro (WaveMetrics, Portland, OR, USA), WinASCD (Guy Droogmans, KU Leuven, Belgium), OriginPro (OriginLab, Northhampton, MA, USA) and Prism (Graphad, Jolla, CA, USA) software.

### 4.4. Data Analysis

Analyzing software was used as mentioned above for each method. When statistical analyses were performed, the data were tested for normality and tested for significance with a Student’s *t*-test for normal distributed data or a Mann–Whitney test for not normal distributed data in case of two group comparisons. When n > 2 groups were compared, statistical significance was tested with a one-way ANOVA with a subsequent Tukey’s post hoc test for normal distributed data, a Kruskal–Wallis test with a subsequent Dunn’s post hoc test for not normal distributed data or a Friedman test for repeated measures not normal distributed data. Results were considered significant when *p* < 0.05. Figure 1 was generated using the online tool ChemDraw 19.0.0 (Perkin Elmer Software, Waltham MA, USA) and Inkscape (Inkscape). OriginPro was used for all other data display throughout the manuscript.

## Figures and Tables

**Figure 1 ijms-22-13652-f001:**
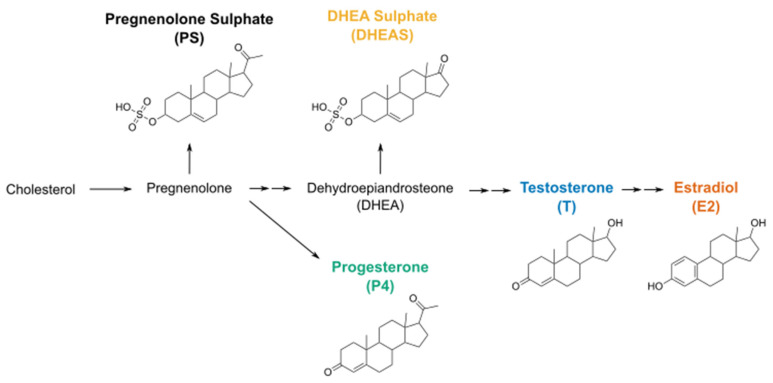
Simplified schematic overview of the steroidogenic pathway and molecular structures of DHEAS, estradiol (E2), progesterone (P4), and testosterone (T).

**Figure 2 ijms-22-13652-f002:**
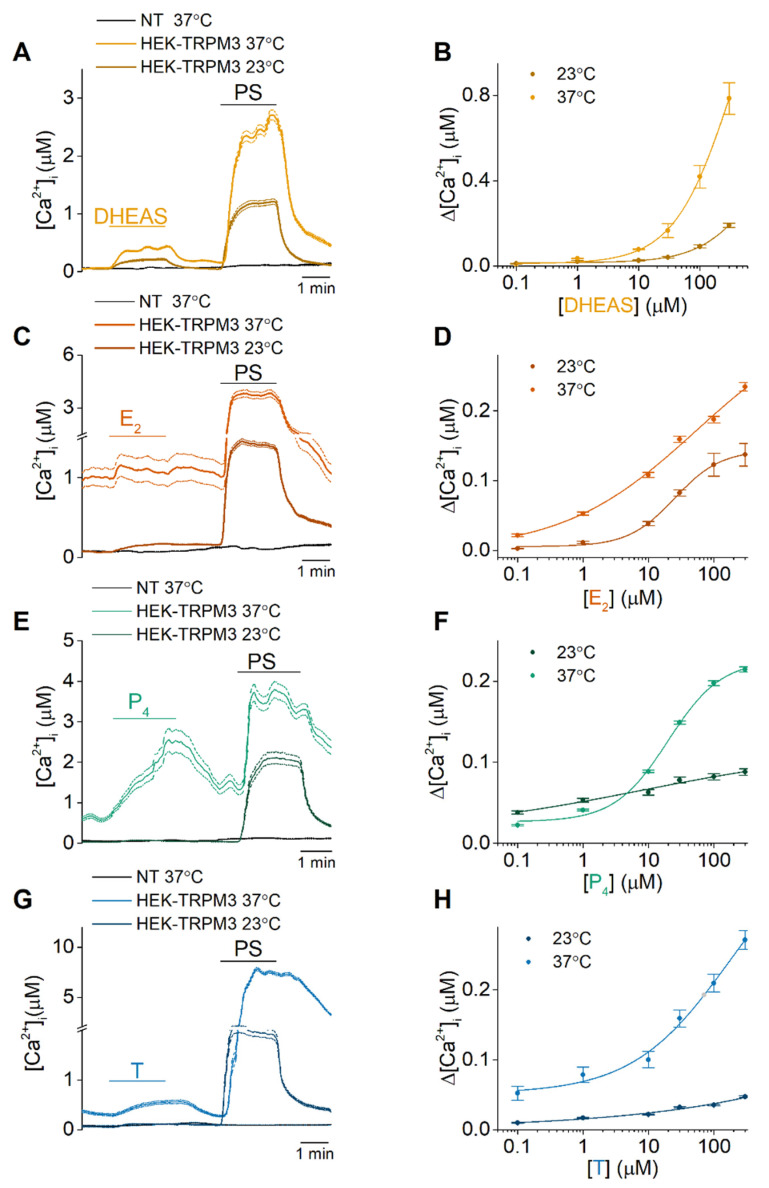
DHEAS, estradiol, progesterone, and testosterone act as weak TRPM3 agonists with increased TRPM3 activation at 37 °C. (**A**,**C**,**E**,**G**) Time course of intracellular calcium concentrations ([Ca^2+^]_i_) of HEK-TRPM3 and nontransfected HEK293T (NT) cells upon application of 300 µM DHEAS, E_2_, P_4_ or T, respectively, followed by 40 µM PS. (**B**,**D**,**F**,**H**) Concentration-response curves of DHEAS (N = 4, n = 1973), E_2_ (N = 4, n = 963), P_4_ (N = 4, n = 602) and T (N = 5, n = 512) at 23 °C (grey) and of DHEAS (N = 3, n = 1457), E_2_ (N = 4, n = 778), P_4_ (N = 4, n = 964), and T (N = 5, n = 485) at 37 °C (colored trace). Data are represented as mean ± SEM.

**Figure 3 ijms-22-13652-f003:**
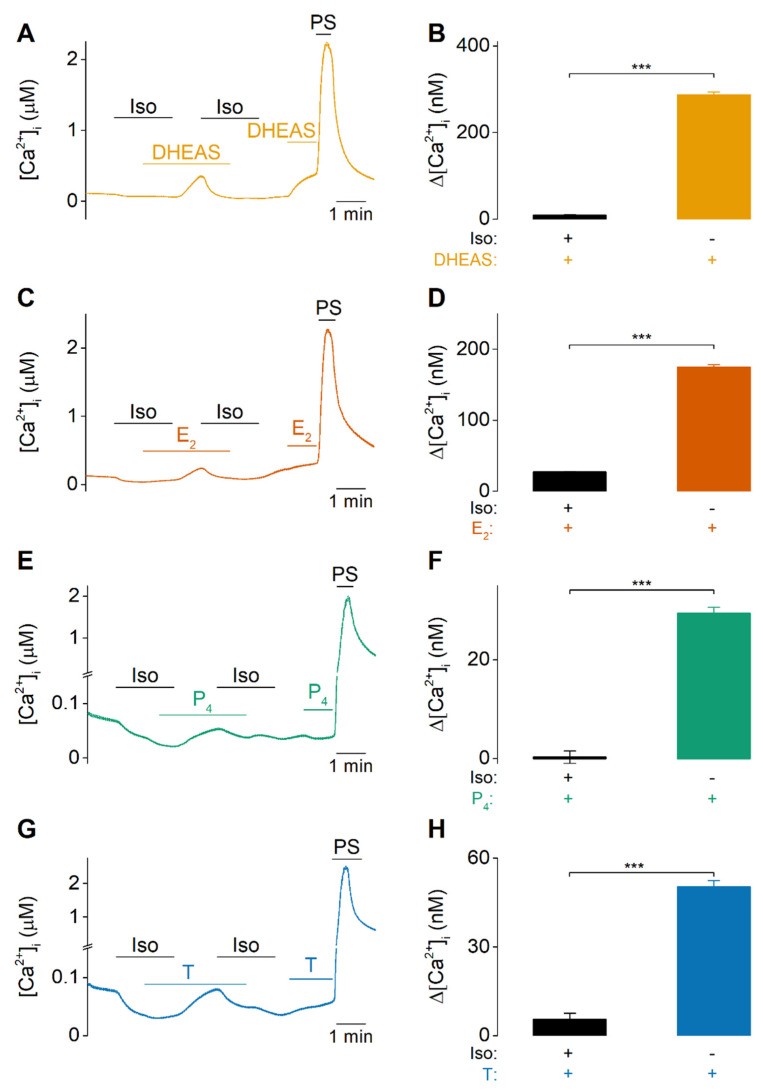
TRPM3-specificity of DHEAS, estradiol, progesterone, and testosterone responses. (**A**,**C**,**E**,**G**) Time course of intracellular calcium concentrations ([Ca^2+^]_i_) of HEK-TRPM3 upon application of 100 µM DHEAS (N = 4, n = 1857), 100 µM estradiol (E_2_) (N = 4, n = 1345), 300 µM progesterone (P_4_) (N = 4, n = 1818) or 300 µM testosterone (T) (N = 4, n = 1730), respectively, in the presence or absence of 5 µM isosakuranetin (Iso). (**B**,**D**,**F**,**H**) Corresponding amplitudes of the calcium imaging experiments in *(***A**,**C**,**E**,**G**). Data are represented as mean ± SEM. Statistical analysis using a Mann-Whitney U test. *p* < 0.001 ***. *p* values for *(***A**,**C**,**E**,**G**): *p* < 1 × 10^−15^.

**Figure 4 ijms-22-13652-f004:**
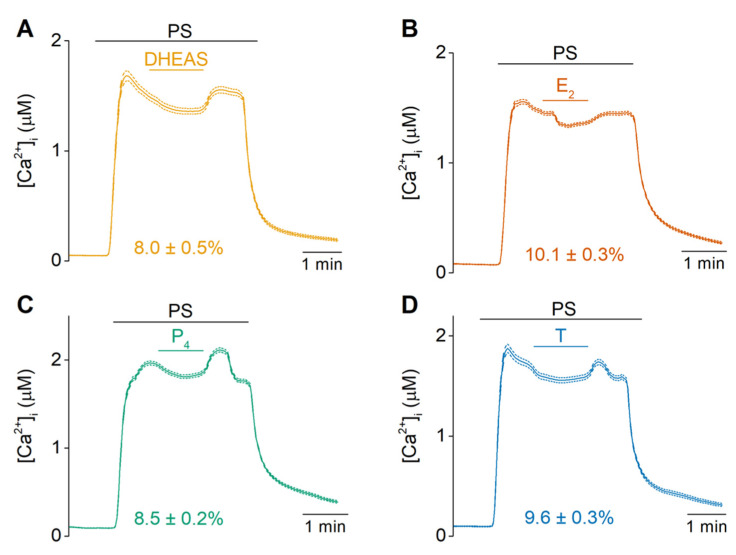
Evidence for competition between PS and DHEAS, estradiol, progesterone or testosterone. (**A**–**D**) Time course of intracellular calcium concentrations ([Ca^2+^]_i_) of HEK-TRPM3 upon application of 40 µM PS, followed by a coadministration with 100 µM DHEAS, E_2_, P_4_ or T, respectively. Competitive antagonism results in a block of PS-response by 8.0 ± 0.5% (N = 2, n = 667), 10.1 ± 0.3% (N = 4, n = 1649), 8.5 ± 0.2% (N = 2, n = 840), and 9.6 ± 0.3% (N = 3, n = 1387), respectively. Data are represented as mean ± SEM.

**Figure 5 ijms-22-13652-f005:**
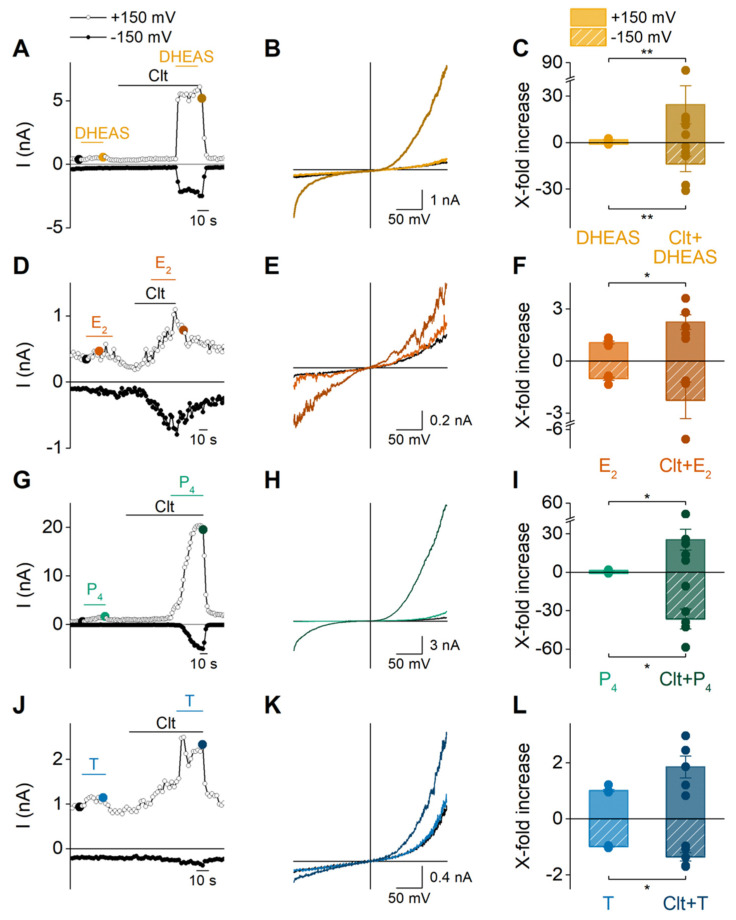
Potentiation of DHEAS, estradiol, progesterone, and testosterone responses by clotrimazole. (**A**,**D**,**G**,**J**) Time course of patch clamp experiments on HEK-TRPM3 cells upon application of 100 µM DHEAS, E_2_, P_4_, and T alone or in combination with 10 µM clotrimazole. All experiments were performed at room temperature. (**B**,**E**,**H**,**K**) The corresponding I-V curve of the indicated points in (**A**,**D**,**G**,**J**) after stimulation with DHEAS, E_2_, P_4_, and T, respectively. (**C**,**F**,**I**,**L**) X-fold potentiation of currents in absence and presence of Clt at −150 mV (shaded) and +150 mV for DHEAS (n = 6), E_2_ (n = 5), P_4_ (n ≥ 5), and T (n ≥ 5), respectively. Data are represented as mean ± SEM. Statistical analysis using two-sample t-tests or Mann–Whitney tests between groups. *p* < 0.05 *, *p* < 0.01 **. *p*-values for (**C**): ±150 mV: *p* = 0.0051; (**F**): +150 mV: *p* = 0.0216 and −150 mV: *p* = 0.0947; (**I**): +150 mV: *p* = 0.0437 and −150 mV: *p* = 0.0105 with Welch correction; (**L**): +150 mV: *p* = 0.1709 and −150 mV: *p* = 0.0358.

**Figure 6 ijms-22-13652-f006:**
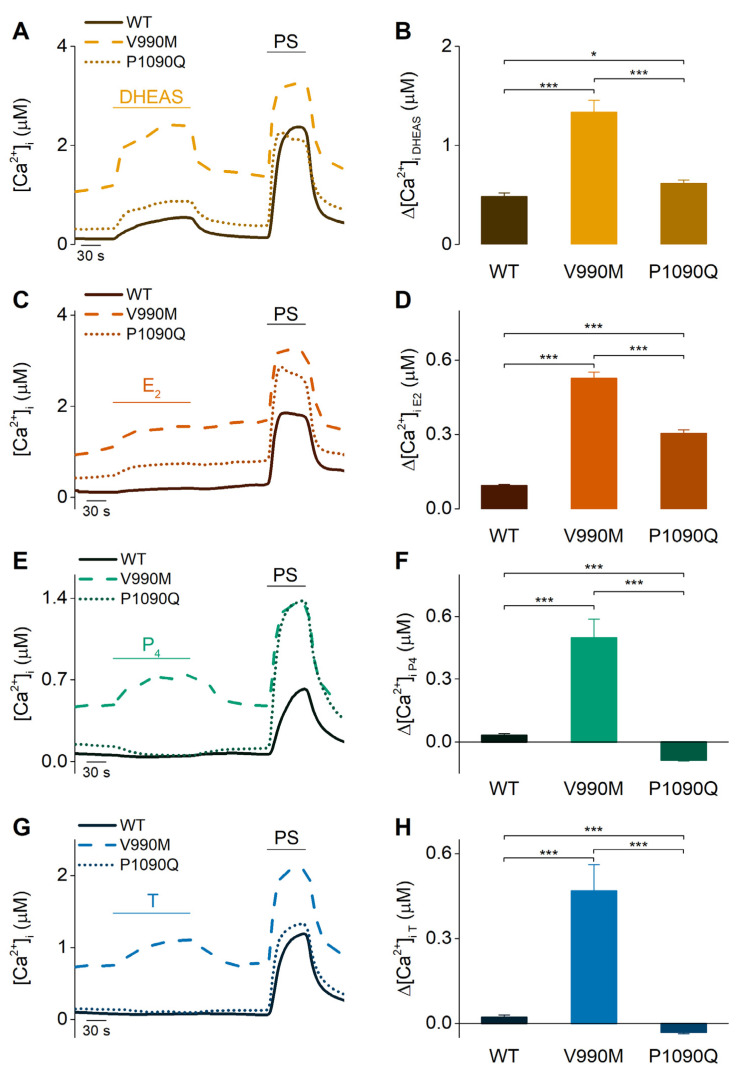
DHEAS, estradiol, progesterone, and testosterone showed altered responses in TRPM3 disease-mutations. (**A**,**C**,**E**,**G**) Time course of intracellular calcium [Ca^2+^]_i_ in wild-type (WT), V990M or P1090Q transfected HEK293T cells upon application of DHEAS (100 µM), E_2_ (100 µM), P_4_ (300 µM) or T (300 µM), respectively. (**B**,**D**,**F**,**H**) Corresponding amplitudes of the calcium imaging experiments (with N = 3 for DHEAS: WT: n = 174, VM: n = 203, PQ: n = 196; N = 4 for E_2_: WT: n = 242, VM: n = 264, PQ: n = 186; N = 3 for P_4_: WT: n = 635, VM: n = 491, PQ: n = 823; N = 3 for T: WT: n = 605, VM: n = 463, PQ: n = 603). Data are represented as mean ± SEM. Statistical analysis using a Kruskal-Wallis test. *p* < 0.05 *, *p* < 0.001 ***. *p*-values for (**B**): WT/VM: *p* < 1 × 10^−15^, WT/PQ: *p* = 0.0122, VM/PQ: *p* = 1.29 × 10^−12^; (**D**): WT/VM and WT/PQ: *p* < 1 × 10^−15^, VM/PQ: *p* = 1.59 × 10^−6^; (**F**): WT/VM: *p* = 1.87 × 10^−13^, WT/PQ and VM/PQ: *p* < 1 × 10^−15^; (**H**): WT/VM, WT/PQ and VM/PQ: *p* < 1 × 10^−15^.

**Figure 7 ijms-22-13652-f007:**
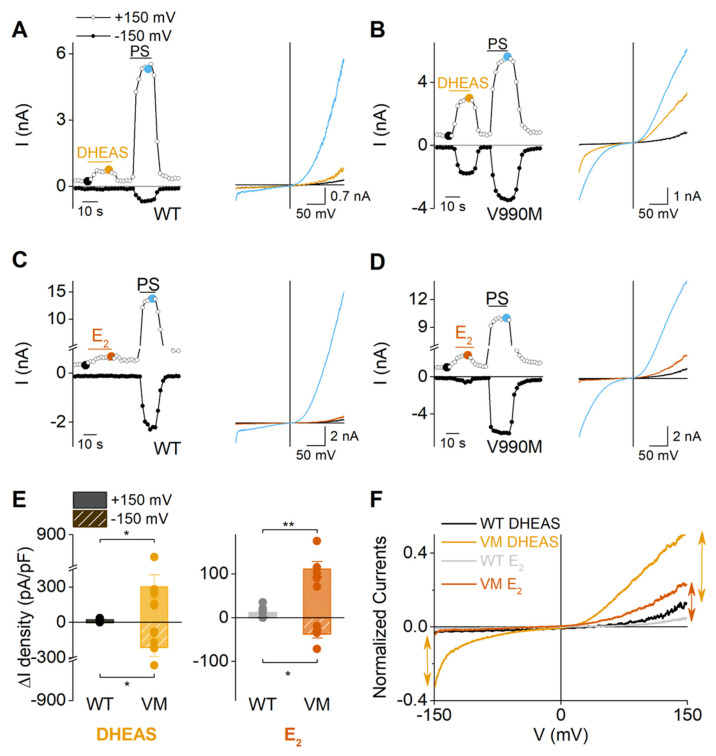
Increased DHEAS and E_2_ sensitivity of the V990M variant compared to WT. (**A**,**B**) Time course of patch clamp experiments on WT or V990M transfected HEK cells, respectively, upon application of 100 µM DHEAS followed by 40 µM PS and the corresponding I-V curve of the indicated points. (**C**,**D**) Time course of patch clamp experiments on WT or V990M transfected HEK293T cells, respectively, upon application of 100 µM E_2_ followed by 40 µM PS and the corresponding I-V curve of the indicated points. (**E**) ΔI densities in WT or V990M transfected cells after DHEAS (n = 5) or E_2_ (n ≥ 5) application at −150 and +150 mV. Data are represented as mean ± SEM. Statistical analysis using a two-sample *t*-test or Mann–Whitney test. *p* < 0.05 *, *p* < 0.01 **, *p*-values for DHEAS: ±150 mV: *p* = 0.0122; E_2_: +150 mV: *p* = 0.0038, −150 mV: *p* = 0.0125. (**F**) IV-curves normalized to the maximal response at +150 mV for each cell. Arrows illustrate the amplitude increases between WT and V990M during DHEAS (yellow) or E_2_ (red) treatment.

## Data Availability

The data presented in this study are available on request from the corresponding author.

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
