# Peer review of "Partial Agonistic Actions of Sex Hormone Steroids on TRPM3 Function"

_ijms, 2021, doi:10.3390/ijms222413652_

Round 1
Reviewer 1 Report
In this manuscript entitled “Partial agonistic actions of sex hormones steroids on TRPM3 function”, the authors assess the agonistic effects of the sex hormones on the TRPM3-trigered ion kinetics. The manuscript also addresses an important links between the TRPM3 disease variants and the sex hormones.
Overall, the manuscript is well-written and informative; however, the authors could strengthen the manuscript by addressing the following concerns.
- In line 297-299, claims that difference in the hormone concentrations create discrepancies are not convincing. Because the TRPM3 overexpression level and distribution are unknown, it is difficult to determine whether or not the appropriate concentration is used. The authors should be more cautious in this conclusion.
- The authors should comment on how the partial agonistic effect contributes to the living body physiologically.
- In the Figure 2BDFH, Figure 3BDFH, Figure 5CFIL, Figure 6BDFH, and Figure 7E, the authors should provide the number of the experiments.
Author Response
Overall, the manuscript is well-written and informative; however, the authors could strengthen the manuscript by addressing the following concerns.
We thank the Reviewer for his/her overall positive assessment of our manuscript as well as his/her comments. In the revised version of the manuscript, we addressed each comment as outlined below.
- In line 297-299, claims that difference in the hormone concentrations create discrepancies are not convincing. Because the TRPM3 overexpression level and distribution are unknown, it is difficult to determine whether or not the appropriate concentration is used. The authors should be more cautious in this conclusion.
We addressed this remark now in lines 310-316, by adding a sentence that discusses the potential difficulties that arise out of the used expression systems.
- The authors should comment on how the partial agonistic effect contributes to the living body physiologically.
The potential physiological role of the partial agonistic effect is now discussed in the revised manuscript (line 319-325). In addition, we further inserted a small discussion on the potential biological impact of the blocking of these neuro-steroids on PS-induced TRPM3 activity (line 331-337). Finally, we finish with a general discussion of physiological processes that could be potentially influenced through steroidal actions on TRPM3 (line 420-430).
- In the Figure 2BDFH, Figure 3BDFH, Figure 5CFIL, Figure 6BDFH, and Figure 7E, the authors should provide the number of the experiments.
We thank the reviewer for his/her attentive reading. We included the experimental numbers and cell numbers in all experiments accordingly. This concerns Figures 2-7 and Figures S2-6 of the revised manuscript. Additionally, we inserted single data points for statistical data recorded in patch clamp experiments (Figures 5, 7 and Figure S5).

Reviewer 2 Report
In the manuscript, Persoons and colleagues found that sex hormones DHEAS, estradiol, progesterone, and testosterone are partial agonists of TRPM3 channel, all these ligands have the same binding site as PS by using calcium imaging and electrophysiology methods. The studies were well planned, and the results are very interesting. I only have some minor comments.
1, It would be better to add a time course of calcium concentration recorded at 23°C for Figure 2A, C, E, G.
2, Figures 2,3,4 used the same PS concentration (40 µM). However, the calcium concentration in response to PS in these three figures were quite different. Were the experiments carried out in same or different conditions? Like different temperature?
3, Figure 4 showed the DHEAS, estradiol, progesterone or testosterone compete bind the PS binding site and reversible blocked the PS-induced calcium response. In Figure 2, with PS continuous present at the end of each time course, there were also some calcium response fluctuations during the PS perfusion. So, it would be necessary to include a calcium concentration time trace with PS only to clarify.
4, Figure 5 missed the voltage of recording for panels A,D,G,J.
5, For I-V curve, it would be better to clarify the recording voltages in methods. It would be necessary to include the recording voltages in all I-V curves of Figures 5 and 7.
6, It would be better to include the statistical results in the results, like t values, F values, degree of freedom and p values.
Author Response
Partial agonistic actions of sex hormone steroids on TRPM3 function
Eleonora Persoons 1,2, Sara Kerselaers 2, Thomas Voets 2, Joris Vriens 1, †,* and Katharina Held 1,2, †
In the manuscript, Persoons and colleagues found that sex hormones DHEAS, estradiol, progesterone, and testosterone are partial agonists of TRPM3 channel, all these ligands have the same binding site as PS by using calcium imaging and electrophysiology methods. The studies were well planned, and the results are very interesting. I only have some minor comments.
We thank the Reviewer for his/her correct summary of our manuscript and the constructive criticism. We addressed each comment as outlined below.
- It would be better to add a time course of calcium concentration recorded at 23°C for Figure 2A, C, E, G.
Following the reviewers’ suggestion, we added a time course recorded at room temperature for Figures 2A, C, E and G in the revised version of the manuscript.
- Figures 2,3,4 used the same PS concentration (40 µM). However, the calcium concentration in response to PS in these three figures were quite different. Were the experiments carried out in same or different conditions? Like different temperature?
We thank the reviewer for this attentive question. Indeed, the experiments were performed in different experimental conditions. In Figure 2, we show time courses of recordings performed at room temperature and at body core temperature (37°C). In Figures 3 and 4 however, we recorded only at room temperature, which explains the lower calcium values upon application of the same concentration of PS. Figures 3 and 4 are in a similar range for the calcium responses to PS. Small variations can occur between the Figures, as these recordings were performed on different days with some possible variability in the expression levels of TRPM3 depending on the passage or density of the cells. However, all experiments within one set of experiments were recorded always at the same day for all steroids, to avoid intra-experimental differences. Consequently, the variability of the PS-induced calcium responses within a single figure are rather constant.
- Figure 4 showed the DHEAS, estradiol, progesterone or testosterone compete bind the PS binding site and reversible blocked the PS-induced calcium response. In Figure 2, with PS continuous present at the end of each time course, there were also some calcium response fluctuations during the PS perfusion. So, it would be necessary to include a calcium concentration time trace with PS only to clarify.
We agree with the suggestion of the reviewer and therefore performed an additional recording in order to abolish any doubts about the blocking effect of the steroids on the PS-induced calcium responses. The new results are included as a supplementary figure (Figure S3) in the revised version of the manuscript (lines 160-162). During the PS application (4 min) the vehicle DMSO (0.1 %) was co-applied. These results show only a very limited effect of the Vehicle application, resulting in a very weak block that is far below the effect of the steroidal compounds.
- Figure 5 missed the voltage of recording for panels A,D,G,J.
We thank the reviewer for his/her attentive reading. We inserted the voltages for panels 5A, D, G and J at the top of Figure 5.
- For I-V curve, it would be better to clarify the recording voltages in methods. It would be necessary to include the recording voltages in all I-V curves of Figures 5 and 7.
We thank the reviewer for this suggestion. The recording protocol is now described in the Materials and Methods section line 485-486 of our revised manuscript.
- It would be better to include the statistical results in the results, like t values, F values, degree of freedom and p values.
We thank the reviewer for this suggestion. We inserted all p-values of our statistical results in the figure legends of our revised manuscript. This concerns Figures 3, 5, 6, 7 and Figures S4-S6. As we always compare small group numbers, we don’t deem it necessary to include degrees of freedom. Likewise, we think that t values and F values would not provide important additional information given that all p-values are provided, and we feel it would overload the manuscript and make it unnecessarily hard for the reader.
